# Evaluation of Different Pressure-Based Foot Contact Event Detection Algorithms across Different Slopes and Speeds

**DOI:** 10.3390/s23052736

**Published:** 2023-03-02

**Authors:** Samuel Blades, Hunter Marriott, Sandra Hundza, Eric C. Honert, Trent Stellingwerff, Marc Klimstra

**Affiliations:** 1School of Exercise Science, Physical & Health Education, University of Victoria, Victoria, BC V8P 5C2, Canada; 2Academy of Sport and Physical Activity, College of Health, Wellbeing and Life Sciences, Sheffield Hallam University, Sheffield S1 1WB, UK; 3Human Performance Laboratory, Department of Kinesiology, University of Calgary, Calgary, AB T2N 1N4, Canada; 4Canadian Sport Institute Pacific, Victoria, BC V9E 2C5, Canada

**Keywords:** running, gait analysis, pressure, event detection, algorithms, wearable technology, smart insoles

## Abstract

If validated, in-shoe pressure measuring technology allows for the field-based quantification of running gait, including kinematic and kinetic measures. Different algorithmic methods have been proposed to determine foot contact events from in-shoe pressure insole systems, however, these methods have not been evaluated for accuracy, reliability against a gold standard using running data across different slopes, and speeds. Using data from a plantar pressure measurement system, seven different foot contact event detection algorithms based on pressure signals (pressure sum) were compared to vertical ground reaction force data collected from a force instrumented treadmill. Subjects ran on level ground at 2.6, 3.0, 3.4, and 3.8 m/s, six degrees (10.5%) inclined at 2.6, 2.8, and 3.0 m/s, and six degrees declined at 2.6, 2.8, 3.0, and 3.4 m/s. The best performing foot contact event detection algorithm showed maximal mean absolute errors of only 1.0 ms and 5.2 ms for foot contact and foot off, respectively, on level grade, when compared to a 40 N ascending and descending force threshold from the force treadmill data. Additionally, this algorithm was unaffected by grade and had similar levels of errors across all grades.

## 1. Introduction

Recent advancements in running wearable technologies have enabled continuous monitoring of running mechanics in field-based settings. Accurate running wearables could provide an essential tool for evaluating injuries and providing measurements that can be utilized to improve performances [1,2,3]. The quantification of running gait with wearable technology requires accurate identification of foot contact events (FCEs), such as initial foot contact (IC) and toe-off (TO). FCEs are important landmarks in the running gait cycle that allow for the determination of temporal metrics, such as stride rate (SR), ground contact time (GCT), and swing time (ST). Further, FCEs are needed to correctly segment phases of the gait cycle and allow appropriate kinematic and kinetic comparisons between strides, limbs, and across cohorts [4,5]. The accurate detection of FCEs in wearable devices is dependent on both the properties of the sensors being used and the algorithm used to process the sensor signals [6,7,8]. However, before wearable running sensors can be used to quantify running gait, the accuracy of such technologies should be assessed relative to laboratory-grade measurement systems.

The current laboratory-based method of determining FCEs employed by gait researchers is through the use of vertical ground reaction force (vGRF), and is typically measured using in-ground force plates or force measuring treadmills [4,8,9,10,11,12]. The standard approach for FCE identification from measured force is to determine when the force has risen above (IC) and fallen below (TO) a specific force threshold (ex. 20 N) [8,10]. While this is a standard and simple approach, the detection of FCEs in running from vGRF requires the use of specialized lab-based equipment. Consequentially, there is limited research on runners in their normal training or competition environments [4,7]. To address these limitations researchers have investigated different wearable sensors for their ability to accurately detect FCEs during running, including the use of accelerometers, gyroscopes, force sensing resistors (FSRs), and in-shoe plantar pressure measurement systems (PPMSs) [1,6,7,12,13]. Among these technologies, in-shoe PPMSs have the unique advantage of providing both kinematic and kinetic running gait data, including the path of the center of pressure and the distribution of forces under the foot. Additionally, PPMSs have been shown to be more accurate in the detection of FCEs when compared against inertial sensors in walking trials [13]. While good agreement has been shown in the detection of FCEs between PPMSs and those determined from lab-based vGRF in walking trials [4,11,13], few researchers have investigated the accuracy of in-shoe PPMSs in the detection of FCEs during running [1,12,14]. As running has faster loading rates and shorter contact times compared to walking, this presents a different set of constraints and challenges for the development of pressure-based running wearable technology [6]. Additionally, running presents a broad range of use-case conditions, such as surface, grade, speed, and individual differences in foot strike patterns, all of which may impact the ability of a wearable to accurately determine FCEs [15]. Moreover, the potential changing properties of a PPMS, such as drift, creep, or hysteresis, which can occur over extended duration use [14,16,17,18], could further impact a given FCE detection algorithm’s functionality and accuracy. Taken together, these considerations have required researchers investigating the use of pressure-based sensors to quantify running gait to employ different algorithms in an effort to approximate the accuracy of the standard force plate or force treadmill approaches more closely. For example, early research by Hausdorff et al. [11] evaluated an algorithm for the detection of running FCEs using inexpensive FSR data which were shown to be highly accurate for IC detection (mean 0 ms ± 3 ms,) and TO detection (mean −1 ms ± 8 ms) when compared to an in-ground force plate. However, their algorithm processed uncalibrated (voltage) sensor data which have a nonlinear response and may not function similarly on calibrated (linear) pressure sensor data. Additionally, their algorithm was only tested on level grade running and thus, may not be appropriate for use in a running wearable which could be used across a broad range of conditions. More recently, Mann et al. [7] conducted an investigation evaluating the accuracy of a custom in-shoe pressure insole system. Mann et al. [7] employed a FCE detection algorithm similar to Hausdorff et al. [11]. However, their testing was similarly only conducted on level grades at self selected speeds (2.78–3.33 m/s). Harle et al. [6] evaluated custom built hardware and algorithms to detect FCEs from in-shoe pressure data during sprinting. Their algorithm employed a different method than Hausdorff et al. [11] and was shown to be accurate for the detection of FCEs in sprinting. However, their algorithm was not tested at recreational running paces, or on different grades, and so may not be appropriate for a running wearable device. While these present interesting and valuable approaches by researchers there are also many other event detection algorithms from other signal processing methods that may support the appropriate determination of FCEs using PPMSs. For example, Zhou and Zang [19] developed a novel signal onset detection method, that could be employed when detecting FCEs from in-shoe pressure data where signal onsets may not be as abrupt as force signals.

The evaluation of multiple FCE methods may help advance the development of pressure-based running wearables. Despite this, there are no investigations to date, that evaluate the accuracy of different algorithms used to determine FCEs during running when measured from PPMSs. Additionally, there are, to date, no investigations that have evaluated the accuracy of different pressure based FCE algorithms, across different running speeds and grades. Thus, the purpose of this study was to (a) evaluate the accuracy of different pressure based FCE detection algorithms across different running speeds and grades to (b) evaluate which foot contact event detection algorithm has the best agreement with a gold standard method.

## 2. Materials and Methods

### 2.1. Participants and Protocol

A total of 18 (9 male, 9 female) participants were recruited aged 19–40 years (mean: 28 ± 5 years). Participant height ranged from 1.55 to 1.93 m (mean: 1.73 ± 0.10 m) and body mass ranged from 52.0 to 87.5 kg (mean: 66.6 ± 10.3 kg) [20]. All participants were free from injury at the time of testing and were familiar with treadmill running. The protocol and methodology were approved by the University of Calgary human research ethics committee and all participants provided written informed consent before participating. All participants wore their normal running footwear which spanned several brands. Prior to testing, each participant was fitted with the Pedar system (100 Hz, Novel, Munich, DEU) where the sensors were placed within the participant’s shoe on top of the existing insole (Figure 1). Each Pedar insert contains 99 pressure sensing elements. The Pedar system has a pressure sensing range of 15–600 kPa and has been shown to be valid under laboratory conditions at estimating vGRF [21,22], and to have excellent pressure measurement accuracy [16,18].

Following a self-selected warm-up, participants ran on a force-instrumented treadmill (2400 Hz, Bertec, Columbus, OH, USA) while vGRF and plantar pressure were simultaneously recorded. Subjects ran on level ground at 2.6 m/s (9.4 km/h; 6:24 min/km), 3.0 m/s (10.8 km/h; 5:33 min/km), 3.4 m/s (12.2 km/h; 4:54 min/km), and 3.8 m/s (13.7 km/h; 4:23 min/km), six degrees inclined at 2.6, 2.8, and 3.0 m/s, and six degrees declined at 2.6, 2.8, 3.0, and 3.4 m/s. Data were collected for 75 s. At the start of each trial, subjects were asked to perform a stationary stance on the treadmill belt followed by three consecutive two-foot jumps. The jumps were used during post hoc analysis to synchronize data from the two systems. Following the jumps, the belt speed was increased to the selected running speed for a given trial. Data were cropped for analysis from 25 s onward in the trial to ensure only steady state running was collected.

### 2.2. Post-Hoc Data Processing

All post-hoc data processing (Figure 2) was performed using custom software (LabVIEW™ 2018 National Instruments™, Austin, TX, USA). To facilitate direct comparisons between FCEs determined from the reference vGRF signal and FCEs determined from in-shoe pressure data, both the force and pressure signals were resampled to 1000 Hz [6,23]. The vGRF data were then filtered using a zero-lag, 50 Hz, fourth order, low-pass Butterworth filter. To increase the generalizability of the results of this investigation, a single pressure sum signal which comprised a sum of all 99 pressure sensors was generated [24]. The sum of pressures (P_sum_) was selected for application on the FCE algorithms, as it constituted a simplified signal that could be easily generated from any PPMS regardless of its configuration (array or discrete) or pressure sensor count [6,7,11]. Additionally, P_sum_ was selected as a signal which is potentially less susceptible to differences in foot strike patterns, and one that closely approximated vGRF signals in profile [6,24]. Thus, for this investigation, the FCEs, as measured from the plantar pressure data, were derived from the P_sum_ signal. Next, the vGRF and the P_sum_ data were temporally aligned based on the synchronizing jumps using cross-correlation [20]. P_sum_ was then normalized [0–100] based on the maximal value within each trial [6]. Finally, the data were cropped to the 50 s period for each trial.

#### 2.2.1. Reference FCE Detection

For each trial, IC and TO event locations were derived from the vGRF signal from the force instrumented treadmill, to provide ‘gold-standard’ reference dataset by which the different FCE detection algorithms were evaluated (Figure 2). A standard threshold crossing method [23,25] was used for FCE detection, with a 40 N threshold for all trials (Figure 3). The 40 N threshold value was determined through iterative testing, to be the lowest threshold to correctly identify FCEs across all speeds, treadmill grades, and participants, where signal artifacts, such as noise due to belt/platform vibration, and shear loading on positive and negative grades were not impacting FCE detection [23,25]. Although other researchers have employed thresholds ranging from 10–50 Ns on force instrumented treadmills for FCE detection, due to the inclines used in this protocol, and the belt vibrations, 40 N was deemed to be the lowest threshold value that could be used across all grades, speeds, and participants without needing to apply additional filtering to the vGRF signal which has been shown to impact the timing of FCE detection [8,23,26].

#### 2.2.2. Pressure Based FCE Detection

The following seven different algorithms were applied to the P_sum_ signal (see Appendix A for the link to a repository containing a custom Python (version 3.6) implementation of each FCE algorithm). Wherever appropriate, the specific parameters used in each of the following algorithms were taken directly from the literature. Otherwise, parameters were iteratively modified and tested until the greatest number of FCEs were successfully detected. Additionally, for each algorithm, a ‘wait’ period was used, such that any FCE detected in the following 20 samples after the first detected IC or TO were removed to eliminate false positives (see Appendix A for addition details on the wait function).

##### FCE Algorithm 1 (FCE1) 2 Threshold Crossing

This algorithm (Figure 4) identifies the indices where the input signal crosses a threshold in an ascending direction and checks to see if the following 20 indices remain above the threshold to find the locations of IC. This has the advantage of eliminating false positives in noisy data where the widths of peaks are typically much shorter in cycle length than the dominant signal [4]. The algorithm also identifies the indices where the signal crosses the same threshold value, in the descending direction, and then checks to see if the following 20 indices were also below the threshold to find the location of TO. For this investigation, a threshold of 10% of the maximum signal was used. This threshold was also determined using iterative testing where the 10% threshold was chosen as the lowest possible threshold for both IC and TO that generated the greatest number of correct FCE detections across all participants and conditions.

##### Algorithm 2 (FCE2) 2 Different Thresholds

This algorithm (Figure 5) uses the same method as FCE1, but uses different thresholds for IC detection and TO detection. This algorithm was included in this investigation to assess if using different static thresholds for IC and TO can account for the potential differences in pressure onset and offset slopes present in some FSR and PPMS signals [6]. Due to the fast rate of signal onset in running, the detection of IC is less sensitive to the selected threshold value. However, near toe off, pressure signals can have a much slower rate of offloading and thus the chosen threshold could have a greater impact on TO detection. While lower thresholds should increase the accuracy of the algorithm, too low of a threshold could leave the algorithm susceptible to changes in signal, such as drift [27], or residual pressure present during the swing phase. For this investigation, the IC threshold was set at 5% of the maximum signal and the TO threshold was set to 10% of the maximum signal Figure 5). The thresholds of 5 and 10% were selected in a similar manner, as described for FCE1.

##### Algorithm 3 (FCE3) 2 Peak Derivative

This algorithm (Figure 6) takes the first derivative of the P_sum_ signal [13]. The signal derivative is then filtered at 12 Hz using a zero-lag low-pass Butterworth filter. A peak detect function is applied on the signal derivative to find the locations of the signal peaks which are then assigned to the locations of IC. The peak detect function is then applied to the inverse of the signal derivative to determine the locations of the negative peaks as the locations of TO.

##### Algorithm 4 (FCE4) 2 Slope Extension Method

To find the location of IC, this algorithm generates a linear function based on the magnitude and location of the maximal positive slope (the positive peak of the first derivative) of the signal and finds the time intercept where the pressure signal would be zero. To find the location of TO, this algorithm (Figure 7) generates a second linear function based on the magnitude and location of the maximal negative slope (the negative peak of the first derivative), and finds the time-intercept where pressure would be zero. This algorithm does not rely on static values and is responsive to different rates of signal onset and offset, which may increase the reliability of this algorithm when used in PPMSs with drifting signals.

##### Algorithm 5 (FCE5) 2 Low-Frequency Unity

This algorithm (Figure 8) uses a fourth order, 2 Hz, zero lag, low-pass Butterworth filter to generate a highly smoothed sinusoidal version of the P_sum_ signal. A peak detect is then used to find the locations of the peaks and valleys of the smoothed signal. The original P_sum_ signal is then broken into segments of ascending (from valley to peak) and descending (from peak to valley) based on the locations of the peaks and valleys of the 2 Hz filtered signal. A unity line (which is a linear ramp of values going from the start of each segment to the end) is generated. Then the absolute difference between the original signal and its unity line is calculated and the location of the maximal difference is determined to be the location of IC from the ascending segments and TO for the descending segments [19]. Similar to FCE4, this algorithm also does not rely on static values, potentially increasing its reliability regardless of running technique, surface, or grade.

##### Algorithm 6 (FCE6) 2 Harle et al. 

This algorithm (Figure 9) is based on the method of foot contact event detection presented by Harle et al. [6]. A rough estimate of IC and FO events is first found using a threshold crossing (FCE1) with a threshold of 50% of P_sum_ maximum signal. This provides a late estimate of IC and an early estimate of TO locations. Following this, the first derivative of the input signal is generated. Next, a fine estimate of IC is found using a search window from the derivative signal that is 10 samples backwards from the rough IC location. The algorithm then searches within that window for the last index with a value less than 0.3, as the fine estimate of IC. A fine estimate of TO, is similarly created using a refined search window from the inverse of the derivative signal which has 10 samples going forward from the coarse estimate TO event. The algorithm then searches this window for the first value that goes below 0.3 of the derivative signal as the fine estimate of TO.

##### Algorithm 7 (FCE7) 2 Mann et al. & Hausdorff et al.

This algorithm (Figure 10) is based on the method of foot contact event detection presented by Mann et al. and Hausdorff et al. [7,11]. First, a coarse estimate of IC and TO are determined using the FCE1 algorithm using a threshold based on the mean of P_sum_ signal. The first derivative of the P_sum_ signal is then filtered using a fourth order, 12 Hz, zero lag, low-pass Butterworth filter. Similar to the method presented in FCE6, a search window from the derivative signal that is 30 samples backwards from the rough IC location is generated. IC is defined as the time point within the search window when the first-grade derivative diverged from the zero line but remained below 1. Similar to IC, TO was determined using a search window from the derivative signal that is 30 samples forwards from the rough TO location. Within this search window, TO was defined as the time point when the first-grade derivative converged from a negative value of 1 towards the zero line.

### 2.3. Data Analyses and Statistics

Each of the 7 FCE algorithms were used to determine locations of IC and TO events for every step in every trial. Additionally, for each set of FCEs detected, the stance time (GCT) was also calculated as the time (ms) between a given IC and its successive TO event. Error values were then taken as the absolute value of the difference between the vGRF based IC, TO, and GCT values and those determined from each of the pressure based FCE detection algorithms.

For each trial, the mean absolute error (MAE) was calculated between the foot contact events, as detected by the reference vGRF signal and the FCE detected by each of the pressure based FCE algorithms. All statistical analyses were completed using JASP™ version 0.16.4. The statistical significance was accepted as *p* < 0.05.

#### 2.3.1. Algorithms across Speed (Level Grade)

To assess the differences between each algorithm across speeds (2.6, 3.0, 3.4, and 3.8 m/s), a 4 (speed) by 7 (algorithms) repeated measures ANOVA was performed on the MAE for IC, TO, and GCT. Tukey’s HSD post-hoc tests were used in the case of significant main effects and interactions. For safety reasons, the fastest running speed (3.8 m/s) was only completed on level treadmill grades. Thus, for the algorithm by speed part of the investigation, only level grade data were used.

#### 2.3.2. Algorithms across Speed, across Grades

To assess the differences between each algorithm across speeds and grades, a 2 (speed) by 3 (grades) repeated measures ANOVA was performed on the MAE for IC, TO, and GCT. Only 2 speeds were run across all 3 grades and therefore the analysis was limited to only 2.6 and 3.0 m/s. Tukey’s HSD post hoc tests were used in the case of significant main effects and interactions.

## 3. Results

The IC, TO, and GCT determined using a 40 N threshold on the vGRF data were used as the reference criterion against which each pressure based FCE algorithm was assessed. Absolute differences from the reference criterion value were calculated and the mean values for each trial termed mean absolute error (MAE) measured in milliseconds. Descriptive statistics for each algorithm for IC, TO, and GCT are summarized in Table 1.

### 3.1. Algorithms across Speeds (Level Grade)

For the IC MAE, there was a significant main effect for algorithm (Figure 11). Post hoc revealed that FCE3 and FCE7 had a significantly greater MAE than all the other algorithms. Additionally, FCE1 had a significantly smaller MAE than all the other algorithms. For the TO MAE, there was a significant main effect for algorithm and a speed by algorithm interaction. Post hoc revealed that FCE3 had a significantly greater MAE than all other algorithms and FCE* had a significantly smaller MAE than FCE3 and FCE7. Additionally, FCE3 at 2.6 m/s had a significantly larger MAE than FCE3 at all other speeds. For the GCT MAE, there was a significant main effect for algorithm and a speed by algorithm interaction. Post hoc revealed that FCE3 was significantly different than all other algorithms and FCE3 was different across all speeds, with the lower speeds having a greater error. Furthermore, algorithm FCE7 was different than all other algorithms but not different across speeds.

### 3.2. Algorithm across Speed and across Grades

For the IC MAE, there was a significant main effect for grade and significant speed by grade and grade by algorithm interactions (Figure 12). Post hoc revealed that for grade, downhill and level are different than uphill. Post-hoc revealed that for grade by algorithm interaction for FCE3 and FCE4, downhill and level were different than uphill. For the TO MAE, there was a significant main effect for algorithm and a speed by algorithm interaction. Post hoc revealed that all other algorithms were different from FCE3 and FCE7. There were no other differences. For GCT, there was a significant main effect for algorithms and no other interactions. Post hoc revealed that algorithms FCE1, FCE2, and FCE4 were significantly lower than all other algorithms. Furthermore, algorithm FCE3 had a significantly higher MAE than all other algorithms.

## 4. Discussion

This is the first study to compare different algorithms used to determine FCEs from PPMSs against a gold-standard instrumented treadmill during running on different grades. Overall, the results suggest that while many algorithms performed well against the standard vGRF FCE approach, and were consistent across speeds and grades, some common algorithms performed poorly against the standard and were speed and grade dependent. These results support the use of a few valid and reliable PPMS FCE algorithms that may be useful in research and smart sensor applications.

When vGRF is not available for gait research, foot switch sensors, technologically similar in principle to PPMSs, have been used as a proxy for determining FCEs [26]. Indeed, these types of pressure sensors have been the standard against which kinematic FCE algorithms are compared, across different running speeds, grades, and foot strike styles [28]. This demonstrates the importance of ensuring that the most accurate PPMS FCE algorithm can be established, not only for smart sensor applications, but also as a tool for running research. To ensure that their foot switch technology and algorithm was valid for comparison of different kinematic-based FCE algorithms, Alvim et al. [26] performed a small pilot study with two male subjects walking and running at different speeds. Alvim et al. [26] compared footswitch-derived FCEs to a force platform using the technique presented by Hausdorff et al. [11] (FCE7 in this study). During the running assessment, they found approximately ±10 ms error for IC and a ±18 ms error for FO for rearfoot strike pattern and approximately ±40 ms error for IC and TO in the midfoot strike pattern which caused them to exclude midfoot strikes from their comparison. In the present study, this same algorithm (FCE7) was one of the poorest performing algorithms and had between 25 ms MAE for IC and 20 ms MAE for TO, which together result in approximately 40 ms MAE in GCT. While the present study did not quantify foot strike pattern, these findings are consistent with the pilot study by Alvim et al. [26] and suggest that other algorithms may be more suitable alternatives to the one developed by Hausdorff et al. as standard FCE PPMS algorithms when force plates are not available.

In the present study, when comparing each algorithm across speeds on level ground, a few algorithms were identified as poor performers against the reference. The peak derivative algorithm (FCE3) had the highest MAE for IC, TO, and GCT time. Further, FCE3 was dependent on speed with a worse performance at lower speeds. The Hausdorff and Mann algorithm (FCE7) had a higher MAE at IC than all other FCE algorithms. This resulted in FCE3 and FCE7 performing the worst on the GCT MAE. FCE3 uses the peak positive derivative to determine IC and the peak negative derivative to determine FO. Based on the nature of rising and falling signals, such as the force or pressure during walking and running, it is expected that the peak positive slope (derivate) would occur later than an onset of force or pressure and a peak negative slope (derivate) would occur before the complete removal of force or pressure. Therefore, it is not surprising that FCE3 performed as it did. Additionally, the speed dependent response for this algorithm is also expected. For example, as speeds increase and the rate of force/pressure development and removal increases, the time between the initial increase and decrease in pressure and the positive and negative peak derivatives diminishes. This could explain the improved response of the peak derivative as speed increases. It is important to mention that while FCE3 performed poorly compared to the other algorithms at determining TO and IC events, the peak derivative signal is an important basis for other FCE algorithms tested in this study (FCE 4,6,7), as it can reliably determine the peak rising and falling slope. How the peak derivative signal is used within each algorithm can therefore determine the overall algorithm accuracy, as this can result in poor (FCE 7) or good (FCE 4,6) algorithm performance.

There were a few well performing FCE algorithms in this study, as FCE 1, 2, 4, 5, and 6 all performed well with a low GCT MAE error (10 ms [FCE1] to 20 ms [FCE6]). Interestingly, two of the top performing FCE algorithms were FCE1 and FCE2, which are simple threshold-based algorithms. This result replicates the finding of Hanlon et al. [13] when comparing FSR to force during walking. These two algorithms were consistent across speeds and grades and FCE1 had the least MAE for IC. This result may be related to the fact that the PPMS sensor used in this study is a calibrated capacitive PPMS, as opposed to uncalibrated piezoresistive sensors, such as the footswitch technology. These different types of pressure sensor technologies have different characteristic responses which may require different algorithms for accurate FCE detection [16]. For example, the specific device used in this study has been shown to have high validity compared to a force instrumented treadmill for aspects, such as loading rate, which could support comparable IC detection using similar thresholding techniques [24]. Other PPMS devices may have different characteristic linearity and hysteresis in both static and dynamic conditions, which may change how different FCE algorithms perform with different PPMS sensor makes and types [16]. Additionally, another important consideration is that the algorithms in this study were applied to a pressure sum of all the plantar pressure data. As different PPMSs have different sensing areas and number of active sensors, it would be valuable to compare FCE algorithms applied to discrete or summed pressure data. Taken together, PPMS FCE comparisons may need to be reassessed and determined specifically for the intended PPMS technology.

An important consideration for determining the suitability of an FCE algorithm for smart sensor deployment in running applications is how an algorithm performs across different speeds and grades. In this investigation, it was found that there was a difference across grades for the IC MAE, such that decline and level had a lower MAE than incline for FCE3 and FCE4. As described above, FCE3 is solely based on the peak derivative. In this specific case, the peak derivative on the ascending limb of the pressure occurred later than the rise in force. As the incline on the treadmill may modify the direction of the resultant force, this may result in a decreased vGRF and a longer delay between the force threshold and the peak derivative of pressure. This is also corroborated by the similar result for grade observed in FCE4 which relies upon both the magnitude and timing of the peak derivative slope. As the MAE is also greater in the incline for FCE4, this would suggest a decreased slope consistent with a potentially modified vGRF. Interestingly, there were no other differences across grades and speeds for TO or GCT supporting robust performance of all algorithms. It should also be notes that the treadmill speeds used in this investigation do not include the typical training and competition speeds of elite distance runners [29]. Consequently, additional algorithm testing may be necessary before the results of this investigation can be extended to this athlete population.

When evaluating the algorithms in this study it is important to consider their potential application within wearable sensor technology. An important factor related to this is the processing requirements for accurate determination of FCEs. As the simplest processing intensive algorithms, the threshold-based FCE1 and FCE2, are rather appealing, however important technical limitations in wearable sensor technology may limit their use. For example, FCE1 and FCE2 rely on static values. However, if extensive sensor creep happens throughout a trial, which is common in some PPMS systems [16,18], or over the lifetime of the sensor, then algorithms, such as FCE1 and FCE2, may likely result in unreliable values. Additionally, as PPMS signals are not immune to erroneous data or spikes that can result in false positives for IC or TO event detection, it is common that filtering may be required as a first step in processing. However, the amount of smoothing can potentially negatively impact FCE detection accuracy and may modify the thresholds chosen within specific algorithms [8]. Additionally, filtering may not be viable in embedded environments where processing power and power consumption are heavily constrained.

## 5. Conclusions

This study evaluated the accuracy of different pressure based FCE detection, when compared to a standard laboratory-based method using vGRF from a force instrumented treadmill during running. A few valid and reliable PPMS FCE algorithms, such as simple threshold crossing, have been shown to be valid across various speeds and grades. Such methods may be useful in research and in the validation of smart sensor applications. Despite their accuracy, caution should be used when employing such algorithms in other pressure based wearables due to their known sensor properties, such as drift and hysteresis [16,30], and future research should be employed to evaluate the algorithms in their intended technology application under varied conditions and constraints.

## Figures and Tables

**Figure 1 sensors-23-02736-f001:**
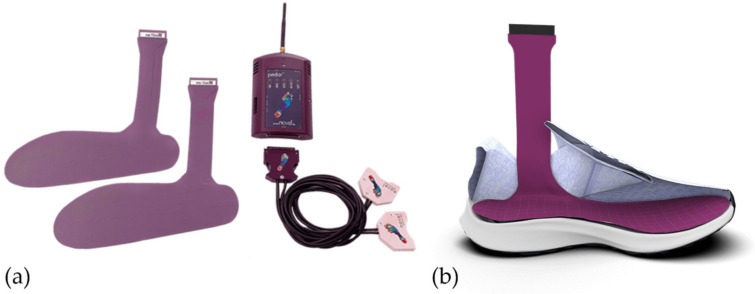
Novel^®^ Pedar™ pressure insole system. (**a**) Complete system (**b**) Pedar insole positioned within an insole (taken from https://www.novel.de/products/pedar/ accessed on 15 January 2023).

**Figure 2 sensors-23-02736-f002:**
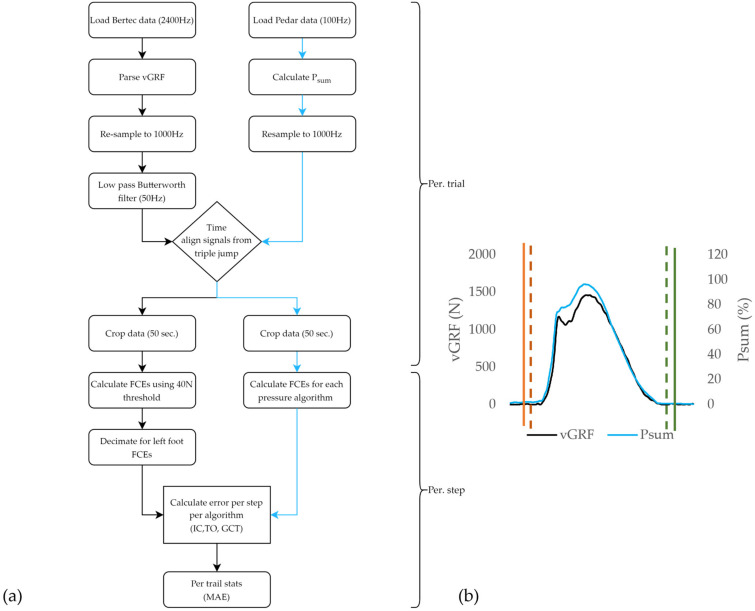
(**a**) Data processing (N = 18) flowchart detailing the processing of the reference vGRF data from the force instrumented treadmill (black) and P_sum_ data from the PPMSs (blue) for each trial and for each stance within a given trial. (**b**) Single stance with overlayed vGRF signal (black) and P_sum_ (blue). Overlayed reference of vGRF initial contact events (red dashed) and reference vGRF toe off events (green dashed) FCEs, as determined using the P_sum_ signal using a given FCE algorithm (solid orange for initial contact and solid green for toe off).

**Figure 3 sensors-23-02736-f003:**
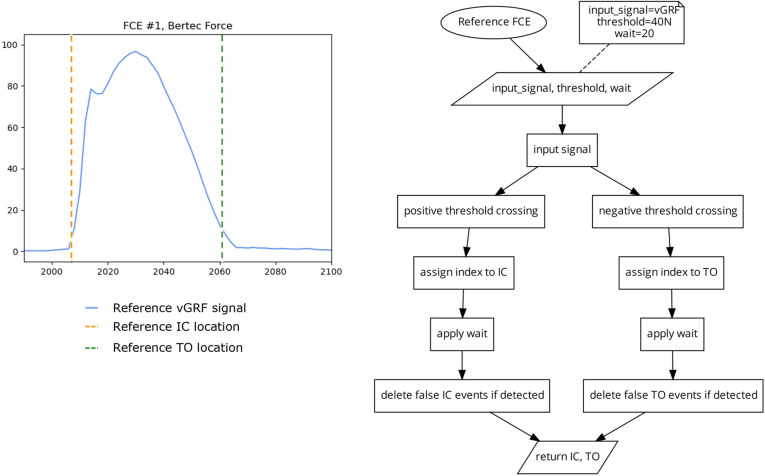
Plot (**left**) displaying a single stride and location of the reference IC and TO events (dashed vertical), as determined by a standard threshold crossing algorithm (**right**) using a 40 N threshold on the vGRF signal.

**Figure 4 sensors-23-02736-f004:**
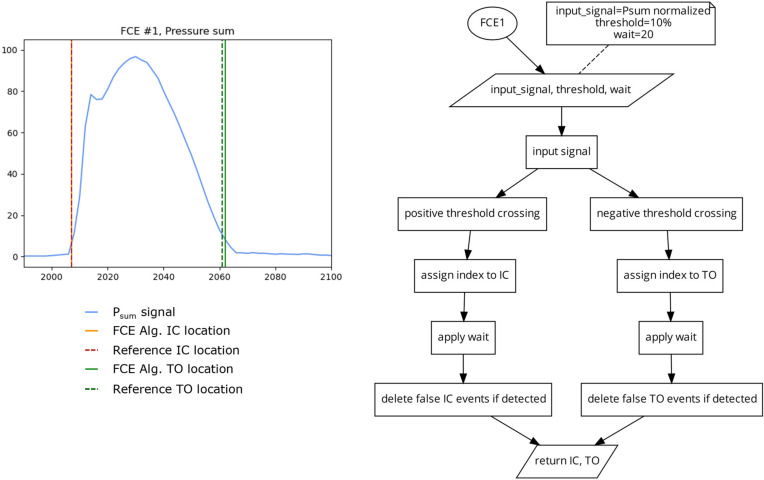
Plot (**left**) displaying a single stride and the location of the reference FCEs (dashed vertical lines) and the FCEs (solid vertical) determined by the FCE1 algorithm (**right**) using a 10% of the maximum signal threshold on the P_sum_ signal.

**Figure 5 sensors-23-02736-f005:**
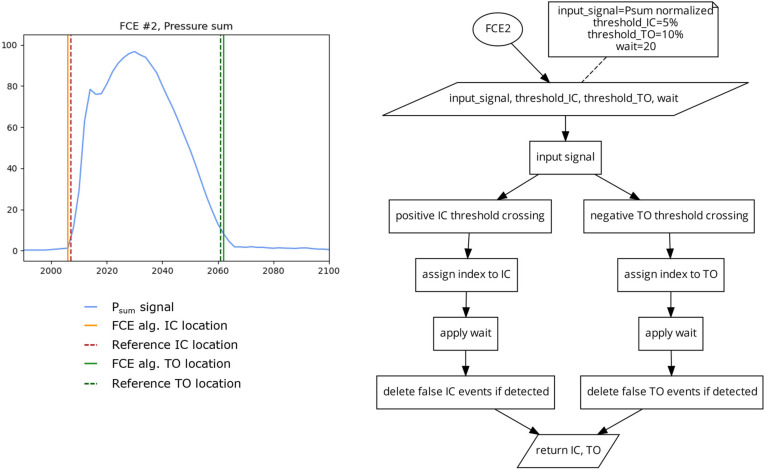
Plot (**left**) displaying a single stride and the location of the reference FCEs (dashed vertical) and the FCEs (solid vertical) determined by the FCE2 algorithm (**right**) using a 5% of the maximum signal threshold for IC and a 10% of the maximum signal threshold for TO.

**Figure 6 sensors-23-02736-f006:**
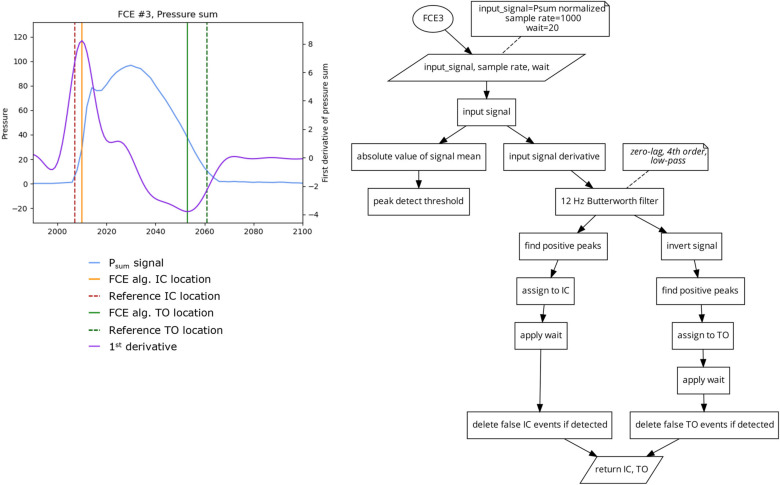
Plot (**left**) displaying a single stride and the location of the reference FCEs (dashed vertical) and the FCEs (solid vertical), as determined by the FCE3 algorithm (**right**).

**Figure 7 sensors-23-02736-f007:**
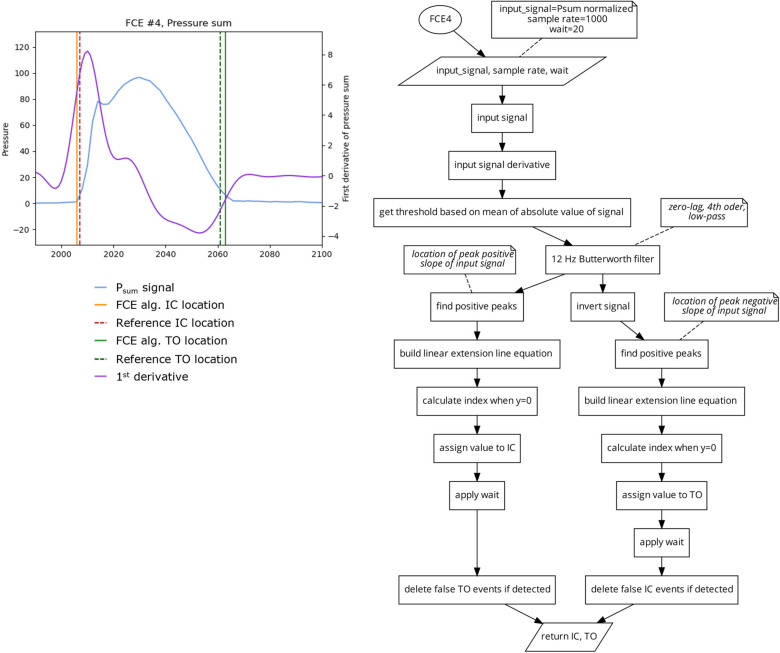
Plot (**left**) displaying a single stride and the location of the reference FCEs (dashed vertical) and the FCEs (solid vertical), as determined by the FCE4 algorithm (**right**).

**Figure 8 sensors-23-02736-f008:**
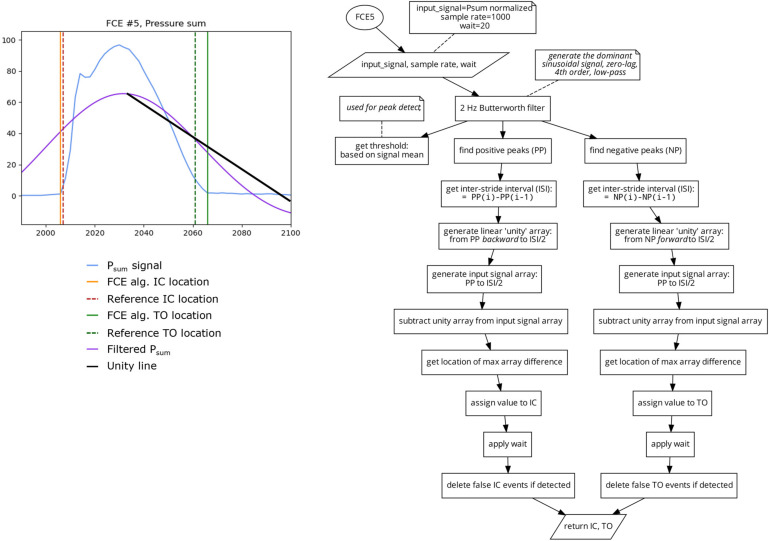
Plot (**left**) displaying a single stride and the location of the reference FCEs (dashed vertical) and the FCEs (solid vertical), as determined by the FCE5 algorithm (**right**).

**Figure 9 sensors-23-02736-f009:**
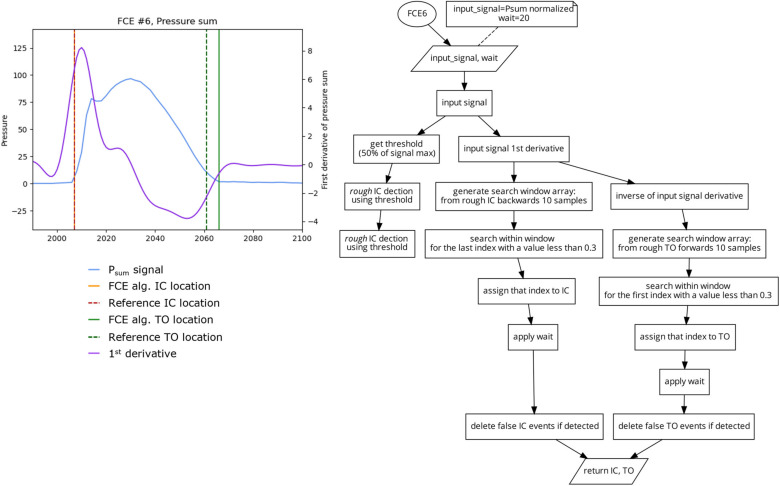
Plot (**left**) displaying a single stride and the location of the reference FCEs (dashed vertical) and the FCEs (solid vertical), as determined by the FCE6 algorithm (**right**).

**Figure 10 sensors-23-02736-f010:**
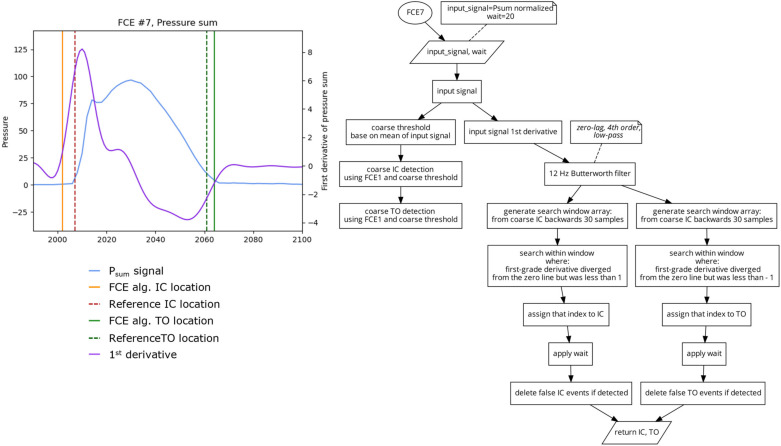
Plot (**left**) displaying a single stride and the location of the reference FCEs (dashed vertical) and the FCEs (solid vertical), as determined by the FCE7 algorithm (**right**).

**Figure 11 sensors-23-02736-f011:**
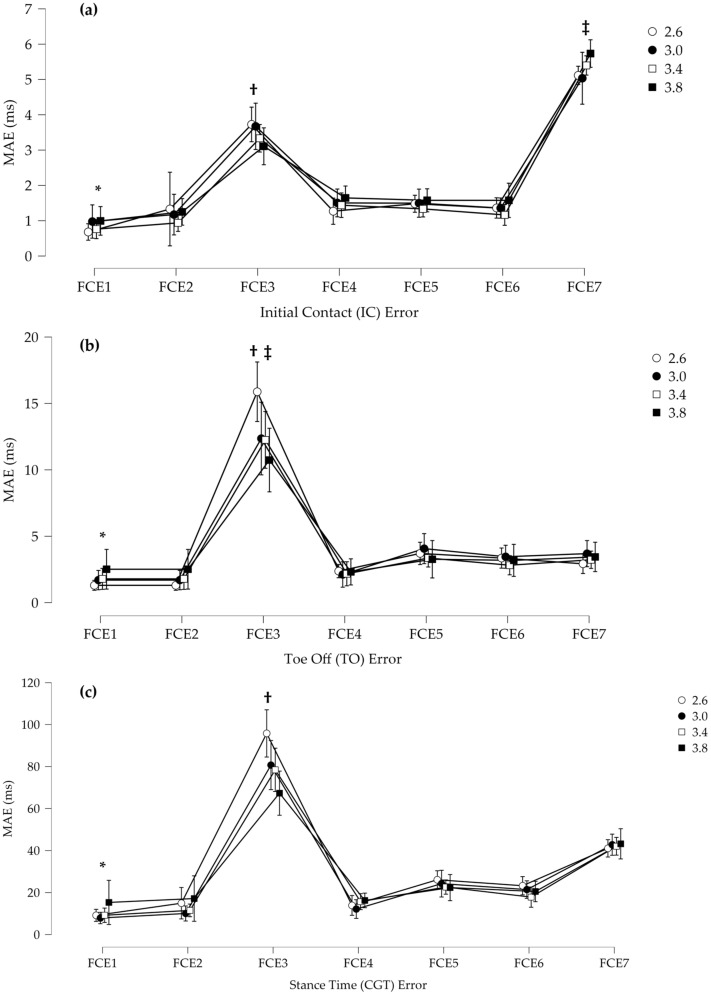
(**a**) Mean absolute error (MAE) for the detection of initial contact (IC). * FCE1 was significantly smaller than all other algorithms, † FCE3 was significantly higher than all others except FCE7, and ‡ FCE7 was significantly higher than all other FCE algorithms. (**b**) MAE for the detection of toe off (TO). * FCE1 had significantly smaller than FCE3 and FCE7, † FCE3 was significantly higher than all other algorithms, and ‡ FCE3 was significantly higher at 2.6 m/s than all other speeds. (**c**) MAE for stance time (GCT). * FCE3 was significantly higher than all other algorithms by speed, and † FCE7 was significantly higher except for FCE3 than all other algorithms. Error bars represent standard deviations.

**Figure 12 sensors-23-02736-f012:**
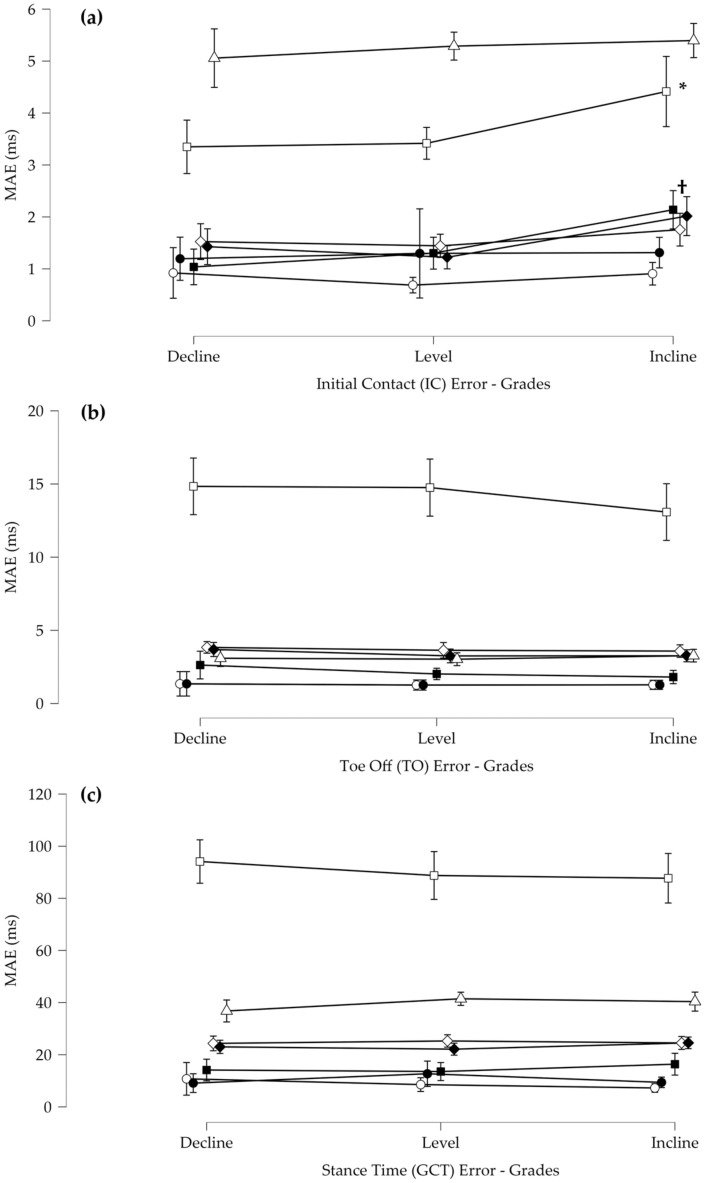
(**a**) Mean absolute error (MAE) for the initial contact events (IC) across all grades (* FCE3 was significantly higher on inclines than all other grades, and † indicates that FCE4 showed as significantly higher on inclines than at all other grades. (**b**) Mean absolute error (MAE) for the toe off events (TO) across all grades (**c**) Mean absolute error (MAE) for stance time (GCT) across all grades. Error bars represent standard deviations.

**Table 1 sensors-23-02736-t001:** Mean and standard deviations of the MAE (ms) for IC, FO, and GCT determined for each algorithm across all speeds (2.6, 3.0, 3.4, and 3.8 m/s) on a level grade.

		Initial Contact (IC)	Toe Off (TO)	Stance Time (GCT)
Speeds	Algorithms	Mean MAE ± SD (ms)	Mean MAE ± SD (ms)	Mean MAE ± SD (ms)
2.6	FCE1	0.7 ± 0.3 *	1.3 ± 0.9 *	7 ± 5
	FCE2	1.3 ± 2.0	1.3 ± 0.9	12 ± 11
	FCE3	3.7 ± 0.9 †	15.9 ± 4.5 †‡	94 ± 24 *
	FCE4	1.3 ± 0.8	2.4 ± 0.9	12 ± 8
	FCE5	1.5 ± 0.5	3.7 ± 1.4	26 ± 8
	FCE6	1.4 ± 0.6	3.4 ± 1.2	24 ± 7
	FCE7	5.1 ± 0.5 ‡	2.9 ± 1.1	41 ± 7 †
3.0	FCE1	1.0 ± 1.1 *	1.7 ± 1.4 *	8 ± 5
	FCE2	1.2 ± 1.2	1.7 ± 1.4	10 ± 6
	FCE3	3.7 ± 1.4 †	12.4 ± 5.5 †	79 ± 25 *
	FCE4	1.5 ± 0.8	2.1 ± 1.6	13 ± 6
	FCE5	1.5 ± 0.8	4.1 ± 2	26 ± 10
	FCE6	1.4 ± 0.6	3.5 ± 1.4	22 ± 6
	FCE7	5.0 ± 1.2 ‡	3.7 ± 1.7	43 ± 9†
3.4	FCE1	1.0 ± 1.1 *	1.7 ± 1.4 *	8 ± 5
	FCE2	1.2 ± 1.2	1.7 ± 1.4	10 ± 6
	FCE3	3.7 ± 1.4 †	12.4 ± 5.5 †	79 ± 25 *
	FCE4	1.5 ± 0.8	2.1 ± 1.6	13 ± 6
	FCE5	1.5 ± 0.8	4.1 ± 2	26 ± 10
	FCE6	1.4 ± 0.6	3.5 ± 1.4	22 ± 6
	FCE7	5.0 ± 1.2 ‡	3.7 ± 1.7	43 ± 9 †
3.8	FCE1	1.0 ± 0.9 *	2.5 ± 3.3 *	10 ± 8
	FCE2	1.3 ± 0.8	2.5 ± 3.3	12 ± 7
	FCE3	3.1 ± 1.0 †	10.7 ± 4.6 †	66 ± 22 *
	FCE4	1.6 ± 0.8	2.3 ± 2.2	16 ± 8
	FCE5	1.6 ± 0.7	3.3 ± 2.9	23 ± 12
	FCE6	1.6 ± 1.0	3.2 ± 2.7	20 ± 11
	FCE7	5.7 ± 0.8 ‡	3.4 ± 2.3	45 ± 12 †

*, †, ‡ indicates where significant differences were found.

## Data Availability

Not applicable.

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
