# Peer review of "Evaluation of Different Pressure-Based Foot Contact Event Detection Algorithms across Different Slopes and Speeds"

_sensors, 2023, doi:10.3390/s23052736_

Round 1

Reviewer 1 Report

This paper evaluated the accuracy of different pressure based FCE detection with comparing to standard laboratory-based method by vGRF from a force instrumented treadmill during running. Some comments are given as following :

1. The introduction about the state of the art should be described clearly.

2. The statements of sections 2.3 and 2.4 are vague and simple,       respectively. The seven algorithms of data processing are given without positive definition. 

3. Figures 3 and 4 are hardly legible and have to be revised.

4. It is better to illustrated the comparisons with compact form to show the

 advantages of this paper.

Author Response

All the best, 

Samuel Blades

Reviewer 2 Report

Peer Review Report

Ms. Ref. No.: sensors-2190735

Title: Evaluation of different pressure-based foot contact event detection algorithms across different slopes and speeds

Authors: Samuel Blades, Hunter Marriott, Sandra Hundza, Eric C. Honert, Trent Stellingwerff, Marc Klimstra

The subject of the article is within scope of the journal. The subject presented in the manuscript is very interesting and the results are very promising. However, the manuscript needs to be improved in my opinion. I recommend the paper for major revision. In particular, the authors should improve the way of presenting the method and results. I believe that the authors will find below some suggestions, which will help them to improve their manuscript (I hope so):

Major comments:

1)      The literature review in the Introduction should be definitively extended.

2)      Why only 18 participants took part in the study. Is it statistically significant?

3)      The authors should present also the formulas for the parameters they used in the current study.

4)      Furthermore, the authors should illustrate somehow the method of obtaining the parameters.

5)      Titles of figures 3 and 4 look like descriptions, which should be rather included in the text of the manuscript. Please shorten the titles and the rest put in the text of manuscript.

Minor comments:

6)      Do not introduce abbreviation in the Abstract of the paper. The authors introduced PPMS – never used within the Abstract.

7)      Line 19, 10.5% written with the other font

8)      Lines 22, 95, 107, 108, 109, 121, 130, 135, 167 (and many others): a space between the number and unit

9)      The authors use in the text the abbreviation “max”. Please use simply the whole term “maximal”.

10)  It is unnecessary to write each time the term “Description: ”

Conclusion:

The subject of the paper and the manuscript are very interesting. I recommend the manuscript for major revision.

Author Response

All the best, 

Samuel Blades 

Round 2

Reviewer 1 Report

The statements of sections 2.3 is still vague. The seven algorithms of data processing are not defined with positive description.

Reviewer 2 Report

The authors answered all my comments. However, in my opinion, it is not enough that the authors write where the algorithm or parameter is described, because it it a crucial description in the presented manuscript. The authors should described them in more details, including formulas and figures, to present how they work. Currently, the authors present a comparison of something what is very poorly described in their paper. The reader can have an impression that it is simply a comparison of something, not so important what. Therefore, I reccomend to add the detailed description. 

Round 3

Reviewer 1 Report

No more comments.

Reviewer 2 Report

The description of the algorithms is better than in the previous versions, but I would expect rather flowcharts instead of giving the whole codes, in addition, as figures. Furthermore, the quality of Figure 3 should be improved. 
